# Rodent Brain Pathology, Audiogenic Epilepsy

**DOI:** 10.3390/biomedicines9111641

**Published:** 2021-11-08

**Authors:** Irina B. Fedotova, Natalia M. Surina, Georgy M. Nikolaev, Alexandre V. Revishchin, Inga I. Poletaeva

**Affiliations:** 1Department of Biology, Lomonosov Moscow State University, 119234 Moscow, Russia; lzglzg@yandex.ru (I.B.F.); Opera_ghost@Inbox.Ru (N.M.S.); humanoid15@yandex.ru (G.M.N.); 2Institute for Higher Nervous Activity, RAS, 119234 Moscow, Russia; revishchin@mail.ru

**Keywords:** audiogenic epilepsy, rodents, fear reaction, behavior genetics

## Abstract

The review presents data which provides evidence for the internal relationship between the stages of rodent audiogenic seizures and post-ictal catalepsy with the general pattern of animal reaction to the dangerous stimuli and/or situation. The wild run stage of audiogenic seizure fit could be regarded as an intense panic reaction, and this view found support in numerous experimental data. The phenomenon of audiogenic epilepsy probably attracted the attention of physiologists as rodents are extremely sensitive to dangerous sound stimuli. The seizure proneness in this group shares common physiological characteristics and depends on animal genotype. This concept could be the new platform for the study of epileptogenesis mechanisms.

## 1. Introduction

It is widely known that the survival of animals (and of rodents in particular) in the wild depends on their acoustic sensitivity (and on their prompt reaction following danger stimulus). This obvious statement usually escapes the attention of investigators who analyze the rather well-known phenomenon of rodent audiogenic epilepsy (AE). At the onset of loud sound rats, mice, and hamsters of several strains develop typical audiogenic seizures. Such seizure fits develop according to the similar pattern in animals of different species and genotypes [1,2,3,4,5,6] and Figure 1. These fits start as violent running and jumpings, the stage named “wild run” or “clonic run” [7]. There were experimental evidences [8] that this stage of the audiogenic fit is “ambiguous” by its mechanisms; animal reaction reveals the traits of the violent flight reaction (defense behavior in order to escape aversive stimulation), but contains some type of forced movements as well, which are regarded as the signs of seizure process initiation. If the experiment is designed in such a way that during “wild run” an animal is able to escape from the sound source (into a safe place), some animals (but not all) will choose to escape [9]. Antipanic treatment ameliorates the wild run intensity [8]. It is likely that the “polygonal arena” for studying the escape reaction in rats could be of use as the device for further analysis of this ambiguity [10]. The further development of the audiogenic seizure in time is well described elsewhere [11,12,13,14]. Of course, the “wild run” stage in animal with audiogenic-epilepsy is different from that of a normal animal experiencing a panic state. But this difference is determined by differences in the CNS function in such animals. The panic state is induced usually by more complicated stimulus or stimulus situation (i.e., frightening environment, predators, etc.) and the prompt panic reaction help animal to find the safer place. Most animals usually don’t develop “wild-run” in response to a loud sound. Non-AE-prone animals in their majority respond to sound onset by more or less intense startle-reaction. This could signify that sound sensitivity in AE-prone animals is abnormally enhanced. Nevertheless it has some features which resemble panic. The phenotypic similarity in movement patterns and the involvement of the similar brain regions (see below) could be the indications of their “relatedness”.

The same conclusions could be drawn concerning the “similarity” between post-ictal catalepsy in AE-prone animals and the freezing reaction of the normal rodent in response to a threatening situation in which presumably there is no way to escape. The common feature of these two states is the suppression of body movements, although significant differences between these two states do exist as well. For example, specific cataleptic muscle tone had been investigated rather extensively. This state models the important aspects of Parkinson’s symptoms. It is widely known that haloperidol catalepsy develops due to abnormalities in striatal DA-receptors. At the same time the detailed research of this peculiar body muscles state (namely the body postural flexibility/rigidity) are not well known. Freezing is the component of the fear-anxiety reaction and it had been analyzed as a component of this state without special attention to muscle tone state *per se*.

This paper represents the attempt to demonstrate that rodent wild run, clonic, and tonic seizures in response to sound as well as the following catalepsy are the animal’s exaggerated reactions to danger, which adopts the real pathological pattern due to the “constellation” of certain genetic elements. Such genetic anomalies could occur in rodent populations and discovered during laboratory breeding. The respective selection in laboratories increased the penetrance of these traits, so that the AE-prone lines could be created.

In other words, the successful selection of rats, mice and hamsters for high AE seizure intensity means that natural rodent populations harbor their respective genetic variants, although no data on AE seizures in wild populations could be found.

## 2. The General Pattern of AE Seizure Development

At the start of EEG era of neurology (around 1940s) Penfield and Jasper proposed that seizures became generalized as the pathological excitation spreads into the brain stem nuclei (i.e., medulla, pons and midbrain), including reticular formation (which had been described around this time). The brain stem structures are extensively connected to many brain regions and their excitation is the basis for epileptic discharge to become widespread.

The participation of brain stem structures in the genesis of audiogenic clonic-tonic seizures had been demonstrated independently in rats of audiogenic seizure prone strains such as the Krushinsky-Molodkina strain [14], which originated from Wistar population and GEPR (genetically epilepsy prone rats) from the Sprague-Dawley outbred strain. Later, the same pattern of AE seizure had been confirmed in the new strain, Wistar Audiogenic Rats (WAR) [6].

According to an explicit review of C. Faingold [12], based on GEPR data, the extensive firing in inferior colliculi (IC) increases before the seizure develops, which is the indication that this structure is the initial point of AE seizure fit development. The “wild running” expression depends on the activation of superior colliculi (SC) deep layers. The development of AE clonic-tonic seizures depends on the pontine reticular nucleus and periaqueductal gray (PAG), with further spread of excitation to the spinal cord [15]. During post-ictal depression, all areas except the pontine reticular nucleus are quiescent. This hierarchical neuronal network of AE, which determines this pathology, does not involve structures rostral to the midbrain.

The electrophysiological and neurochemical investigations made the serious impact concerning the participation of inferior-superior colliculi and other brain stem structures. The participation of PAG, ventral tegmental area (VTA) and *substantia nigra* as well as of other brain stem nuclei in the development of AE seizure [6,12,16,17] was established. Figure 2 (one of the first EEG records of AE seizure in KM rats) shows the neural excitation in the brain stem with typical lack of epileptic activity in the neocortex. However, it is worth noting that the reactivity of the visual cortex decreased after the seizure (less intense responses to light flashes) in comparison to pre-seizure recording.

## 3. The Neuroanatomical Correlates between AE Seizure Attack and Defense Reactions

### 3.1. Corpora Quardigemina

Inferior and superior colliculi had been recognized as the site of AE fit initiation rather long ago [19]. Among other experimental evidences is the fact that bilateral lesions of IC abolished AE-fit development (see [16]). The onset of acoustic stimulus evokes the typical mammalian “startle” reaction, with IC and PAG mediating this reaction [20]. The review of IC role in this respect was performed by N. Garcia-Cairasco [6]. The initiation and propagation of AE activity relies upon hyperexcitability in the auditory system (and IC in particular) [16,21,22,23]. Thus, as it was mentioned above, the IC is considered to be the crucial structure for the AE-fit “ignition”, and it was shown in mouse and rat AE models and in normal animals as well [24,25]. The activation pattern of different nuclei of IC as the process, which is causally connected with IC initiation, was proved using c-Fos immunoreactivity [13,26,27,28,29,30].

The genetic volumetric differences in IC of GEPR and WAR rat strains in comparison to controls were also demonstrated (with differences between these strains as well) [31].

At the same time, the IC is the structure responsible for the development of the innate defense reaction, at least in rodent brains, which could be expressed either as flight or as freezing [25,32,33,34]. The details were reported; ventral and dorsal IC regions are involved in the expression of the defensive reaction and audiogenic seizures, respectively [33], and aberrant neuronal reactions in the IC-SC system were described [22]. Ferreira-Netto et al. [33] demonstrated that in normal rats, freezing inducing chemical stimulation of these two brain regions activates different brain circuits, and apart from IC they include the forebrain structures.

The experimental data demonstrate that wild run could be the expression (or the -homologous) of the fear-induced flight reaction [2], which is the result of neuronal excitation spreading into lower-level brain stem structures [35] and the spinal cord. The special experiments and respective argumentation made it obvious that panic behavior is a phenomenon which is not similar to the anxiety traits [31,36]. Thus, the flight-like stage (wild run) of the AE fit could be regarded as the pathologically increased panic attack. The naturally occurring panic attacks (provoked in rats by ultrasonic stimulation) are the valid model of the respective human disorder [37]. Of course, panic behavior is the phenomenon which is not homogenous, and this was demonstrated by the effects of experimental interventions into the function of GABAergic and opioid systems in *corpora quarigemina, substantia nigra,* and other structures ([31,38,39,40,41], as the examples). The data obtained also demonstrated that forebrain excitation exert the plausible impact in panic-like behavior of different origin [42]. Thus, the AE-proneness is, presumably, the cause of IC auditory function disorder [43,44].

### 3.2. Brain Stem Structures (PAG in Particular)

The ventrolateral part of PAG (vlPAG) is responsible for mediating the excitation from IC [45] and for generation of the tonic AE stage [3,11,18,45,46]. It was demonstrated in GEPR-9 (in electrophysiological experiments) and confirmed in other AE models. It is worth noting that in cases of human partial epilepsy, these attacks sometimes resemble panic attacks as the patients experience the intense fear.

## 4. The Neurochemical Parallels between the AE Fit and Defense Reactions

*Wild run* (*panic reaction)*. Some considerations in this respect were made above. Thus, the “wild run” stage of AE-seizure fit has the traits which permit to make the parallels with the panic-like animal behavior (namely in non-seizure prone animals). The panic-like behavior, realized as the successive neuronal excitation of IC, SC, PAG and other brain stem nuclei, share the neurochemical “sensitivity” with seizure fits induced by sound and demonstrate the participation of forebrain structures [47].

*Freezing reaction.* In rodents (and not only in rodents, but in other mammals as well) the encounter of fear-inducing situation evokes a freezing reaction (i.e., a normal response) which helps to avoid direct contact with danger. Freezing is expressed as the still posture (i.e., complete absence of body movements). The detailed analysis of this brain state in case of electrical stimulation of different PAG regions convinced authors [48] that there exist “at least four different kinds of freezing with specific neural substrates”. However, the brain stem stimulation creates the excitation pattern, which is different from the “natural” one when animal faces the natural danger. This means that direct comparisons are difficult to perform. As different intensities of IC electrical stimulation induced either freezing or escape reactions, the local infusions of semicarbazide or bicuculline (modulating GABAergic system) also induce these two types of reactions, respectively, demonstrating some type of similarity with AE neurochemical traits [49].

*Catalepsy.* The freezing reactions which are induced either in nature, or as the result of direct brain stimulations could be related to abnormal brain state-catalepsy. The cataleptic state induced by haloperidol (modulating dopamine transmission) was shown to be sensitive to aversive stimulations [50], and this type of cataleptic reactions could be modulated by changes in glutamatergic transmission in IC [51]. This could be regarded as the indication that cataleptic muscle tone pattern has probably some common links with the brain stem defense circuits. The special analysis of post-ictal catalepsy and/or of post sound cataleptic-like states in several rat strains indicated the relationship of AE seizures and this type of catalepsy [52]. The parallels of freezing-catalepsy expression could be partially confirmed by neurochemical and genetic data on pinch-induced catalepsy [53,54,55,56], which was shown to depend on brain tryptophan hydroxylase activity. Pinch-induced catalepsy (demonstrated in rats and mice) is regarded as a change in muscle tone, which resembles and probably is related to the non-responsiveness, developed in young pups when the mother transports them by holding the pup by the nape. The pinch-induced catalepsy in AE-prone rats of KM strain correlated with the intensity of their post-ictal catalepsy [55]. The special selection program for the spontaneous catalepsy from the Wistar population demonstrated several peculiarities in this strain such asthe appearance of “nervous” animals in the population of rats, selected for catalepsy as well as the certain percentage of animals which were AE-prone [54].

*The schematic description of neurochemical substrate of AE seizure.* It was shown rather long ago [24] that injecting metal ions and pyridoxal-5′-phosphate to normal mice made them prone to AE, and this change was accompanied by the elevation of glutamate and aspartate levels and the decrease of gamma-amino butyrate in IC. Thus, this chemical mimicking of AE seizure proved that the excitability level of this structure is connected with AS. The neuronal excitation (in norm and pathology) depends to a large extent on the glutamatergic brain system, which was demonstrated by induction of seizures by Glu-receptors agonists. The normal brain function depends on the balance between glutamate and GABA systems, which is deteriorated in cases of epilepsy. In AE, the misbalance of these neuronal circuits had been demonstrated in IC, SC and PAG [29,57,58,59].

Numerous studies demonstrated the involvement of glutamatergic, GABAergic, opioid, serotonin, and dopamine neurotransmitter systems in expression of reactions which belong to the “defense” domain. They are panic-like run, freezing, startle and AE [28,33,35,60,61,62,63,64,65]. Panicolytic-like effect of BDNF in the rat brain stem was also reported [66]. The hamster AE-prone strain demonstrated behavioral and molecular effects as the reaction to cannabidiol and valproate administration [67].

## 5. Genetics of AE

*The expression of genetic acoustic peculiarities in rat and mouse AE strains.* It is natural to ask question, whether the acoustic sensitivity in AE-prone animals is to blame for the audiogenic epilepsy in rodents. There were no data on KM rats, but other models are more or less investigated in this respect [12,68,69,70] (for details see the review in [71]).

*Acoustic “equipment” in AE animals, single mutations.* The finding that acoustic thresholds in AE-prone rodents were elevated, was common for AE data. Cochlea morphological anomalies, differing in details, were present in almost all studies. It is worthwhile to note the following. The defect in acoustic sensitivity was found in Black Swiss mouse strain, carrying the *jams1* gene, which is responsible for AE. However, this acoustic defect and AE-proneness were shown to be independent traits in these mice [69]. The detailed analysis of cochlear functions in albino Frings mice, namely in their inbred descendants, was also performed [72]. The thresholds of cochlear action potentials of the AE-susceptible RB/1 bg mice were abnormally high, while the AE resistant inbred RB/3 bg mice had normal audiograms of evoked potentials. The F1 hybrid mice were heterotic for cochlear function. This RB/1 bg line showed little age-related cochlear loss, which probably accounts for its robust sensitivity to audiogenic seizures over most of its lifespan. Earlier studies demonstrated that in the susceptible RB line, they demonstrated robust evoked potentials with little or no cochlear microphonic events. The susceptible RB/1 bg mice had well-defined potentials and cochlear microphonic [72]. Thus, we see that of course the acoustic anomalies accompany AE traits. However, they probably develop in parallel (i.e., they are not the cause of this pathology).

Frings mice are also the reliable genetic model of AE susceptibility with the gene mass1 (*monogenic audiogenic seizure-susceptible*, now referred to as Mgr1) being identified as responsible for this pathology. This gene codes for the membrane protein, which does not belong to ion channels “family”. Frings mice display a robust AE, demonstrating sound-induced c-Fos immunoreactivity including the external and dorsal nuclei of the inferior colliculi. The subthreshold acoustic stimulation activates c-Fos immunoreactivity in the central nucleus of the IC [27]. Recently the genetic variant of this gene was found in AE prone WAR strain which is thought to be responsible for AE-proneness of these animals [73]. Homozygous carriers of Scn8a gene mutation in mouse, which affects generalized seizure susceptibility, demonstrated also the typical audiogenic epilepsy with high amplitude EEG signals and c-fos immunohistochemistry labelling in the IC [26,30,59].

*Polygenic inheritance.* The successful selection of rat strains for elevated incidences of AE seizures was the first indication that the AE is the genetically determined trait.

The diallel cross and F2 hybrids analysis [14,71,74,75] of AE-proneness in KM strain demonstrated the recessive polygenic nature inheritance, the data on genetic basis of GEPR and WAR strains is rather sparse [17,76,77,78]. Ribak and Morin [26] and Ribak [59] demonstrated that the IC GABA levels in GEPR were increased in comparison to controls and the number of GABAergic neurons was also atypical with the significantly large number of small GABAergic cells. The anomalies in GABAergic cells in IC and SC were found in KM rats, although the histological pattern of differences between AE-prone and resistant animals was different [73,79]. The IC ERK1/2 kinases activity was shown to be different between KM rats and Wistars [80]. The possible animal model for anticonvulsant mechanism was suggested by Arida et al. [81], namely the spiny rat *Proechimys*
*guyannensis*, in which the decreased seizure proneness to forebrain convulsions was found. Although no data on AE in this species could be found.

*Acoustic startle reaction.* The genetic basis of differences in startle reaction in strains, which differ in AE-proneness and/or in defense reactions, was also shown [25,82,83]. The precise location of the chromosome fragment in mice was indicated, which is responsible for pinch-induced catalepsy [56]. These data are of importance as they signify the plausible ways for further investigation of AE-proneness mechanisms and thus for the origin of epilepsy phenomenon.

*More information on genetic differences in neurochemical indices.* The features which could be interpreted as common within pairs “wild run-panic”, and “catalepsy-freezing” are perceived as distant ones in normal “non-epileptic brain”. Although in cases of KM, GEPR and WAR strains numerous neurochemical anomalies were described which could be due to pathological transition from panic flight into wild run stage and from freezing as a defense reaction into catalepsy. The neurochemical deviations from non-AE normal state which are presumably of genetic origin were found in all three best investigated AE-prone rat strains (KM, GEPR and WAR). The brain monoamine system was analyzed as the first one [84,85,86,87,88,89,90,91,92,93], but deviations in other brain systems, which plausibly could affect behavioral reactions, were also noted (e.g., oxidative stress reactivity and histaminergic brain system) [94,95], as well as early gene expression patterns [96]. These neurochemical deviations were described not only in AE-prone animals; it seems that they are inherent to the “epileptic brain” in general [97,98]. It should be noted that these neurochemical peculiarities lay apart from already described glutamate-GABA misbalance in epilepsy.

*The genetic basis of these deviations could be searched in the early stages of CNS development.* It looks more or less plausible that the disturbances in gene expression patterns, which occur rather early in development, could be responsible for the expression of AE. Such disturbances (mutations) nevertheless permit the embryo to develop into the phenotype, which is still compatible with the embryo survival and future viability of an organism (which, probably, is reduced, as these mutation effects were found in laboratory only). The respective mutations are not yet identified, although several examples which probably fit this line of considerations, are now available. It was found that simultaneous genetic inactivation of three transcriptional factors (*PAR bZip* proteins, connected with circadian rhythms-albumin D-site-binding protein proline, hepatic leukemia factor, and thyrotrophic embryonic factor) induce the audiogenic epilepsy in young mice [99]. The target gene of this transcription factors family is the pyridoxal kinase (*Pdxk*), which catalyzes the conversion of vitamin B12 derivatives into pyridoxal-phosphate. The latter is the coenzyme of many enzymes participating in the neurotransmitter metabolism. It was demonstrated that in the triple mouse KOs the levels of brain pyridoxal-phosphate, serotonin and dopamine were decreased [99].

Mutations of other genes, which are identified as being involved in the early stages of CNS development, could also be possible candidates for AE-type dysfunction. These could be also the *disheveled* genes, as proteins, coded by this gene family, which are the important signaling components of *beta-catenin/Wnt pathway*, are important for developmental processes (cell proliferation and patterning etc.). *Dvl3* (−/−) mice died prenatally with several developmental defects, the detected stereocilia orientation anomaly in the organ of Corti being the example, which attracts attention [100].

The hypothesized abnormal “developmental” genes could induce the CNS anomalies, AE symptoms being among them. Such anomalies could cause the channelopathies in the certain brain regions (or in the brain as a whole), which are crucial for AE expression. As channelopathies (the full list not yet described) are the regular findings in many human epilepsy cases, AE models for respective investigations being of certain clinical value. One should not forget that the rodent brain is especially vulnerable for extreme acoustic stimulation for animals that avoid danger in natural habitats.

The general outline presented in this paper does not consider the mechanisms of so-called audiogenic kindling (i.e., the development of specific myoclonic seizures (type of “jerks”)), which have the forebrain localization and appear after systemic repetitive daily sound exposures [1]. This phenomenon is the result of ascending influences from the (abnormally functioning) brain-stem structures of AE-prone animals. At the same time there are two clear indications that the reverse influences (i.e., from the forebrain to brain stem) also exist, as strong olfactory stimulation resulted in the decrease of AE-seizure intensity [101,102].

## 6. The Experimental Evidences Which Are Not Compatible with the Hypothesis Presented

Of course, the parallels which could be derived between wild run AE stage and panic reaction and between cataleptic traits and behavioral freezing are not “complete” as biological variability exists in physiological processes determining the AE. This “similarity” could not be “ideal”, and the respective evidences presented above are not totally convincing. Some arguments against these parallels could be found, as they concern the metabolic correlates of seizure state. The seizure intensity in human patients and in experimental animals depends on the blood pH values and on the NMDA-receptors’ excitability [103,104]. The acidosis (low pH) was described as occurring during seizure process as well [105,106]. The analysis of the brain stem area “responsible” for respiration function was found to be different in WAR in comparison to controls [107]. Schimitel et al. [108] presented the results of complicated study of CO_2_ and PAG stimulation effects on panic-like behavior in rats with the detailed description of animals’ autonomic reactions. Panic expression increased as the result of sodium lactate and inhalation of 5% CO_2_ in panic-predisposed patients [108]. This inspired authors to analyze this issue in animal model. Exposure to CO_2_ induced some kind of behavioral arousal, but also attenuated PAG-evoked immobility (i.e., the homologue of freezing reaction). This data as well as those, cited above, are in contrast with the CO_2_ seizure attenuating effects demonstrated in KM strain, which were found by L.V. Krushinsky as early as the begin of 1950s, this being one of the first experimental data obtained in this model.

## 7. The General Comparison of Normal and Pathological Reactions in AE Models

The general outline of rodent abnormal reaction as the response to loud sound (“audiogenic seizures”) is presented, which could probably be instructive as the comparison was made with the patterns of normal animal behavioral reactions. One may find that the wild run stage resembles the species-typical (although pathologically exaggerated) flight, wherein an animal aims to avoid imminent danger. Similarly, the drastic change in the muscle tone (catalepsy), occurring after the tonic AE-seizure stage could be regarded as pathologically exaggerated biologically normal freezing reaction. As described above, this resemblance could probably be determined by the close brain stem topographical locations of brain substrates of these two types of behavioral reactions. It could be that this parallelism is a mere coincidence. However, the question of the AE origin stays unanswered. Another hypothesis could be tested as well, namely that such region-specific channelopathies arise in rats and mice as the result of the abnormal pattern of CNS development *per se*, the rodent brain being especially vulnerable for loud sound as these animals rely on sound sensitivity in avoiding danger in natural habitats.

The dysfunction of ion-channels (channelopathy) is known not only for AE [109,110], but for epilepsy and neuromuscular pathology in general [111,112,113,114]. Thus the membrane-potential disturbances look like the most crucial proximate cause of AE and other seizure states.

The complex interconnections between genetic elements and behavior are different in different AE strains. Although one may conclude the existence of hypothetical “common pathway”, namely the mechanism which underlie the seizure state (and AE) and defense behavior. One may also hypothesize the *endophenotype* for audiogenic epilepsy—the specific seizure provoking constellation of genetic and (further) neurochemical events (distortions), which function, probably, at both levels—in peripheral hearing organ and in brain structures. The development of regional specific channelopathies and/or of the misbalance in GABA and glutamate systems (both central and cochlear) could be the possible links which would help to delineate the AE endophenotype. The key components of any identified endophenotype are heritability and stability (state independence). Endophenotype approach is useful as, according to I.I. Gottesman and T.D. Gould [115], “it reduces the complexity of symptoms and multifaceted behaviors”, successful identifying of “units” for analysis being the positive result.

The data presented above in a short way demonstrated that apart from phenotypical similarity of AE seizures there are not many traits common for all AE models explored (non- identical hearing system defects and the pattern of brain neurotransmitter systems misbalance). The data of modern volumetric estimation of brains from WAR and GEPR is also the illustration [31]. The dysfunction patterns of brain neurotransmitter systems in cases of AE not always coincide in different AE genotypes.

One more moment should be mentioned in this respect, namely the attempt of Traub et al. [116] to transfer the notion of “central pattern generator”, CPG (which is used in invertebrate neurobiology and neuroethology) to the domain of circuits, generating vertebrate motor patterns. Authors suggest that the concept of CPG could be useful for analyzing not only normal cortical oscillations, but also the transition of normal activity pattern to epileptiform pathology. The generality of CPG notion could be the helpful theoretical framework for further studies of seizure development.

The final summarizing Table 1 and Table 2 are presented (containing the facts and literature references) which could help to acknowledge the new aspect on AE data presented in this review.

## 8. Conclusions

The literature analysis, presented above, represents an attempt to find the “biological roots” of the peculiar trait, inherent to the series of rodent species, namely the production of intense seizure reaction in response to sound. As these seizures typically start as the wild run and jumps, the first parallel, which crosses the observers mind, is that an animal is eager to run away from the source of the unpleasant sensation. This excitation, being enhanced in certain genotypes, could be followed by motor seizures as this abnormal excitation spreads into the brain stem and further downstream to the motor centers. The parallelism between audiogenic epilepsy and abnormally enhanced escape reaction could be perceived as exotic. However, a range human epilepsy cases (first) and the necessity (second) for the search of epileptogenesis roots (taking the whole brain) is not sufficient to reject this point of view.

## Figures and Tables

**Figure 1 biomedicines-09-01641-f001:**
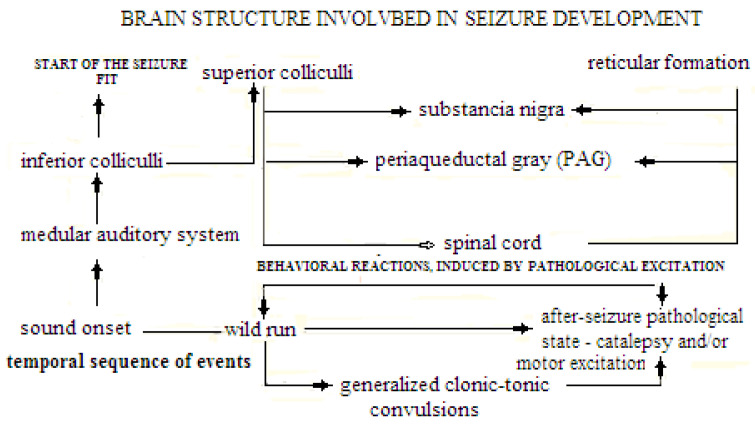
The schematic representation of both-brain structures, involved in the audiogenic seizure progress and the respective animal reactions, which unfold in parallel with certain structure involvement.

**Figure 2 biomedicines-09-01641-f002:**
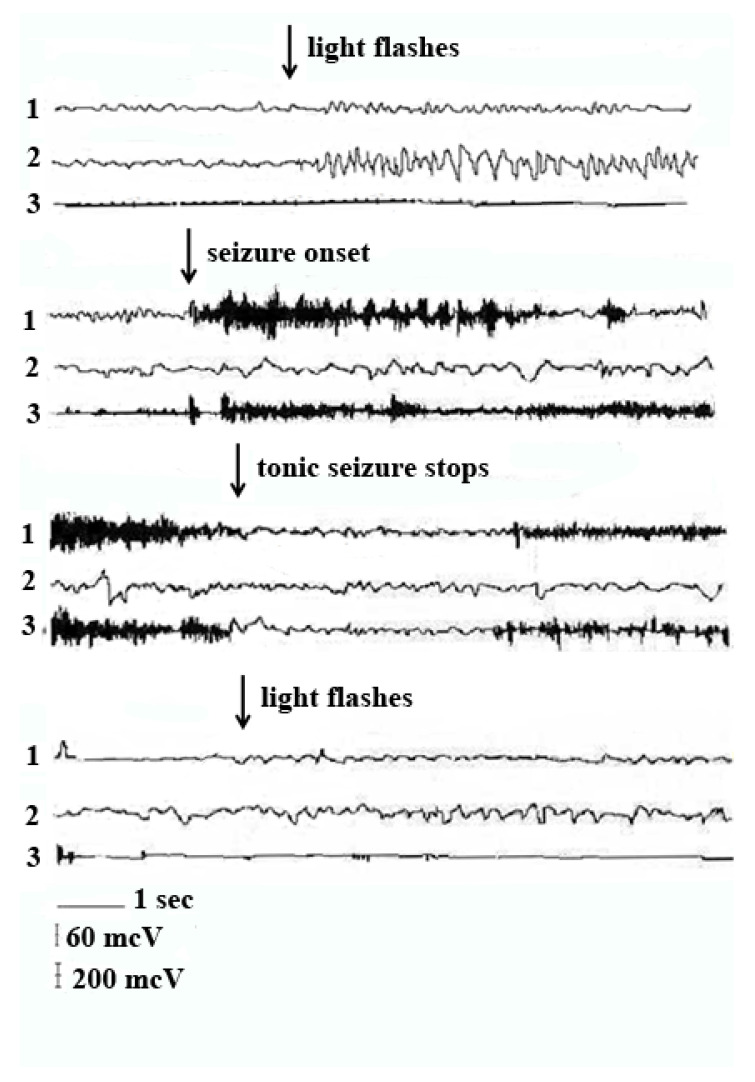
The electroencephalogram (EEG) and electromyogram (EMG) patterns, recorded from chronic electrodes in KM rat during experiment before seizure onset (upper part), during seizure (two middle parts) and after the seizure (the lower part of the figure). Electrodes positions: 1, brain stem; 2, visual cortex; 3, electromyogram (EMG). Calibration: horizontal bar-time scale, vertical bars, 60 mcV for EEG; 200 mcV for EMG (see also [18]).

**Table 1 biomedicines-09-01641-t001:** The data evidencing the neurobiological parallels between defense behavior and AE phenomena.

The Phenomenon Described	The Connection to AE Phenotype	References
Flight behavior and panic reaction	The involvement of colliculi inferior in both states (AE and defense behavior)	[2,25,31,32,33,34,36]
Behavioral freezing	Cataleptic states	[48,49,50,51,52]
The specific neurochemicalstate in brain stem nuclei	The common misbalance in glutamate-GABA signaling	[28,33,35,60,61,62,63,64,65]

**Table 2 biomedicines-09-01641-t002:** The genetic data on AE susceptibility origins.

The Type of Inheritance	The Neurobiological “Unit” Affected	References
Monogenic	Cochlea	[43,68,69,70]
	Non-“channel” coding gene *mass1 (Mgr1*)	[27]
	Different channellopathies	[30,109,110]
	Several genes, expressed in early development, coding *for PAR bZip* proteins, e.g., *Pdxk*. Participant of *wnt* cascade *Dvl3* gene	[114,115]
Polygenic	Dominant-recessive inheritance, diallel cross	[14,17,72,74,75,76,77,78]

## Data Availability

Not applicable.

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
