# Peer review of "Rodent Brain Pathology, Audiogenic Epilepsy"

_biomedicines, 2021, doi:10.3390/biomedicines9111641_

Round 1

Reviewer 1 Report

Manuscript ID: biomedicines-1420167 

Rodent pathology. Auidiogenic Epilepsy

Provide a figure of EEG recordings from an audiogenic seizure and mention the stimulus

10/21

Author Response

Response to reviewer 1

We are eager to express our gratitude to you, as you mention the drawback of the manuscript. We inserted the EEG record from KM rat brainstem, visual cortex and the respective EMG. Thank you. Authors  of the manuscript.

Reviewer 2 Report

The authors review the available literature data about audiogenic seizures (AS). The number of the listed references is respectful. There are three main lines on which the authors discuss the AS. I. Structural correlates: in which the role of some brainstem structures is discussed (mainly the tectum of the midbrain, the PAG and the reticular formation). II. Neurochemical correlates: the roles of the transmitter molecules in the above mentioned brainstem structures. III. Some genetic aspects: gene mutations which are primarily affecting the inner ear.

We indicate that the text is a bit confusing in its present form: a better arrangement of the data could be helpful (e.g.: the arrangement of the data into the three categories - (1) neuroanatomical correlates; (2) neurochemical-biochemical correlates; (3) genetic correlates (mutagenesis).

Better definition of the behavioural reactions is needed, too: we understand that the "wild run" and the "panic reaction" are similar - but still there are main differences between the epileptic animal and the panicking animal. Also, the "freezing" reaction and "catalepsy" should have been defined more clearly. Inserting Tables containing the facts and literature data together could be helpful (e.g.: genetic mutations - cochlea functions - channelopthies).

A major structural revision could be helpful in presenting the data in this review.

Author Response

Response to reviewer 2

We wish to express our sincere gratitude for the Reviewer 2, as he pointed to the not-ideal article composition. We tried to adjust the text in order to clarify the main points of this work.

In the renewed version the changes, which were made, are marked by yellow.

We arranged the data available into the three categories – anatomy, neurochemistry and genetics with respective subdivisions. The sections numbers and respective headings were inserted: 3. The neuroanatomical correlates between AE seizure attack and defense reactions” (with subdivisions). 4. The neurochemical parallels between the AE fit and defense reactions (with subdivisions in the text). 5. Genetics of AE (with subdivisions in the text.

As better definition of the behavioural reactions is needed, we add some clarification into the respective part of the Introduction. Of course there are main differences between the epileptic animal and the panicking animal, but there exist the data (which are mentioned in the paper) that the transition from panic into seizure state could be noticed in several cases.

The respective text was added (in the attempt to explain the similarity and differences between respective behavioral states).

Of course the audiogenic-epilepsy prone animal “wild run” stage is different from that of a normal animal experiencing panic state. But this difference is determined by differences in the CNS function of such animals. The panic state is induced usually by more complicated stimulus of stimulus situation (i.e. frightening environment, predator etc which is displayed as the attempts to flee in the safer place). Usually In response to a loud sound such animals  does not develop “wild-run”( non-AE-prone animals respond to sound onset by more or less strong startle-reaction). This could signify that AE-prone animal’s sound sensitivity is distorted in comparison to norm, although the phenotype similarity in movement patterns and the involvement of the similar brain regions could be the indications of their “relatedness”. The same conclusions could be drawn concerning the “similarity’ between post-ictal catalepsy in AE-prone animals and the freezing reaction of the normal rodent in response to situation in which presumably there is no way to escape. The common feature of these two states is their the suppression of body movements, although the differences between these two states do exist. Specific cataleptic muscle tone had been investigated rather extensively as this state model the important aspects of parkinsonian disease and  it develops due to abnormalities in striatal DA-system, although the detailed research of this peculiar body muscles state (namely – the body postural flexibility/rigidity) are yet  not well known. The freezing as the component of fear-anxiety reaction had been analyzed as a component of fear reaction without special attention to muscle tone state per se.

Two tables were created according to recommendations, which, we hope, will help to acquire this complicated material.

Round 2

Reviewer 2 Report

The manuscript has been improved considerably. The new Tables and the electrophysiological recordings are fitting very well. There are minor spelling errors, which will be corrected during the publication process.

Author Response

The reply to Reviewer

According to the notes of the respected Reviewer, we made several corrections in the places of the text, which were indicated.

Complex sentences (page 2, line 89, 102, 104, 105; page 5, line 161, 167; page 17, line 742) are corrected as well as misspelled N’Guemo name.
The remark, which stayed unanswered, is the “spelling error” on the line 5. Is it the requirement to write Lomonossov with the single ”s”,  or with double “ss”? Lomonossov looks more adequate as in Russian there is no sound “z” in the middle of this name.

Nov 3, 2021

Inga Poletaeva